# A Concise Guide to D-Wave Monitoring during Intramedullary Spinal Cord Tumour Surgery

**DOI:** 10.3390/medicina60081242

**Published:** 2024-07-30

**Authors:** Santos Nicolás Zurita Perea, Pablo Abel Alvarez Abut, Kathleen Seidel

**Affiliations:** 1Department of Neurology, Hospital Privado Universitario de Córdoba, Córdoba X5016, Argentina; santosnicolaszuritaperea@gmail.com; 2Department of Neurosurgery, Inselspital, Bern University Hospital, and University of Bern, 3010 Bern, Switzerland; kathleen.seidel@insel.ch

**Keywords:** D-wave, intramedullary spinal cord tumours, intraoperative neurophysiological monitoring, motor evoked potentials

## Abstract

D-waves (also called direct waves) result from the direct activation of fast-conducting, thickly myelinated corticospinal tract (CST) fibres after a single electrical stimulus. During intraoperative neurophysiological monitoring, D-waves are used to assess the long-term motor outcomes of patients undergoing surgery for intramedullary spinal cord tumours, selected cases of intradural extramedullary tumours and surgery for syringomyelia. In the present manuscript, we discuss D-wave monitoring and its role as a tool for monitoring the CST during spinal cord surgery. We describe the neurophysiological background and provide some recommendations for recording and stimulation, as well as possible future perspectives. Further, we introduce the concept of anti D-wave and present an illustrative case with successful recordings.

## 1. Introduction and Historical Background

Somato-sensory evoked potentials (SSEPs) were initially described in the 1970s as a tool to monitor the functional integrity of the spinal cord during surgery for scoliosis [1]. They consist of bioelectrical signals generated by the nervous system in response to electrical stimulation of the peripheral nerves, and they provide information about the functional integrity of the dorsal somatosensory system [2,3]. It soon became evident that this technique might also guide the removal of tumours involving the spinal cord and its surrounding tissues. During the same period, studies demonstrated that SSEPs could predict postoperative dorsal column impairment following spinal tumour surgery [4]. However, postoperative motor deficits have sometimes been reported despite SSEPs being unchanged during surgery [5,6,7]. Thus, before the introduction of motor evoked potentials (MEPs) to intraoperative neurophysiological monitoring (IOM), surgeons had to rely solely on SSEPs providing continuous feedback from the dorsal columns of the spinal cord. MEPs consist of electrical signals generated in response to stimulation of the motor cortex or subcortical motor pathways. They are recorded from the muscles (muscle MEPs) or through an epidural (or subdural) catheter placed on the spinal cord (D-wave) [8,9,10,11,12]. In the surgical context of IOM, MEPs are elicited by electrical stimulation. The emergence of MEP monitoring techniques represented a significant advancement towards safer surgeries.

Deletis and Kothbauer et al. [13,14] refined MEP techniques to monitor the integrity of the spinal cord function during surgery. Their study elaborated the technique for monitoring muscle MEPs as well as epidural MEPs (referred to as D-waves) under general anaesthesia. Through the correlation of clinical outcomes with intraoperative observations, their studies furnished the initial guidance for evaluating these potentials during surgery on intramedullary spinal cord tumours (IMSCTs). Further, they demonstrated the importance of D-wave monitoring in achieving a larger extent of safe tumour resection even in the absence of muscle MEPs. This was linked to more favourable postoperative long-term motor outcomes in those high-risk surgeries. Since then, the importance of simultaneous monitoring of SSEPs, MEPs and D-waves during IMSCT surgery has been confirmed by many studies [8,9,11,15,16].

In 1998, Kothbauer et al. presented a series of 100 patients who underwent IMSCT surgeries guided by IOM with muscle MEPs and D-waves. However, the study did not include a comparison with the group undergoing surgery without IOM [13]. A historical case–control study in 2006 by Sala et al. provided evidence that D-wave monitoring did improve the long-term outcome following IMSCT surgery, especially for intramedullary ependymoma. They compared two groups of patients: 50 patients operated on under IOM guidance (with D-waves and muscle MEPs) and 50 historical controls operated on without IOM. They found that the McCormick grade variation during follow-up was significantly better (mean, +0.28) in the IOM group compared to the historical control group (mean, −16) [16]. In a recent study in 2021, Skrap et al. observed a high rate of gross total resection (GTR) and long-term favourable functional outcomes after the resection of spinal cord ependymoma. Short- and long-term functional outcomes were best reflected by muscle MEPs and D-wave monitoring, respectively [17].

In the present manuscript, we discuss D-wave monitoring and its role as a tool for monitoring the corticospinal tract (CST) during spinal cord surgery. We describe the neurophysiological background and provide some recommendations for recording and stimulation, as well as a possible future perspective.

## 2. Neurophysiological Background

D-waves (also called direct waves) result from direct activation of fast-conducting, thickly myelinated CST fibres after a single electrical stimulus (single-stimulus technique) to the brain. D-waves travel at a conduction velocity of approximately 50 m/s. When the stimulation is performed at the threshold intensity (the minimum stimulation intensity required to elicit a response), it directly activates the superficial subcortical white matter of the brain. Stimuli exceeding this threshold result in increased amplitude and decreased latency of the D-wave by recruiting additional axons and activating deeper subcortical areas. Thus, the amplitude of the D-wave correlates with the number of activated axons, making it a valuable tool for assessing motor pathway integrity during intraoperative monitoring [10,18].

The D-wave signal is an asynaptic evoked potential, and therefore, it is not significantly affected by anaesthesia. As it reflects the first motor neuron, the D-wave is recordable even under muscle relaxation. However, it is crucial to understand that muscle relaxation would prevent the recording of muscle MEPs. Therefore, total intravenous anaesthesia with a short-acting muscle relaxant is recommended for IOM [10]. Nonetheless, it is important to note that artefacts from paraspinal muscles can sometimes be mistaken for D-waves, leading to waveform misinterpretation and potentially incorrect guidance for the surgeon.

Due to the end of the conus medullaris at the level between the first and the second lumbar vertebrae, the monitoring of the D-wave caudal to the spinal level Th10 is very unlikely to be possible for two reasons: first, because the electrode would overlap the cauda equina rather than the spinal cord, and second because most CST fibres have already terminated at more rostral levels. Thus, the amplitude of the D-wave in the lower thoracic cord is rather small [14].

Another important consideration is that the D-wave may not allow complete unequivocal discrimination between the left and right CST fibres. Assessing specific myotomes or muscle groups is also not possible. Thus, adding muscle MEPs is essential to distinguish which side or which segment is predominantly impacted during any alterations of D-wave signals.

MEPs are associated with the motor function of the patient directly after surgery, whereas D-wave monitoring is valuable in predicting long-term motor outcome and functional recovery [17]. In other words, to assess the motor outcome of the patients after surgery, two time points might be important in the clinical and especially oncological context [19]. First, the motor outcome directly after surgery is mostly reflected by preserved muscle MEPs. Second, the potential to recover, which means that the patient wakes up with a motor deficit directly after surgery but recovers to their preoperative motor function over time (transient deficits), might be acceptable in a context of resection of an IMSCT as the patient might undergo rehabilitation. The potential to recover is reflected by a preserved D-wave (even in case muscle MEPs are lost) [13,15,17,20,21].

In conscious humans, a series of later I-waves (indirect waves) with a periodicity of 1.3–2.0 ms follows the D-wave. These I-waves are generated by the activation of frontoparietal oscillatory intracortical circuits, leading to additional discharges of corticomotor neurons. Under anaesthesia, intracortical synapses are suppressed, resulting in intraoperative recordings that predominantly show D-waves. Nevertheless, strong stimuli or light anaesthesia can still evoke one or more I-waves. I-wave recruitment may also occur after the second or third pulse of train stimuli for MEP monitoring. However, the suppression of I-waves by anaesthesia limits their utility as a monitoring tool [22,23,24].

In patients with large IMSCT, the D-wave might be impossible to record even at the beginning of the surgery. This might happen even in patients with preserved muscle MEPs and without significant pre-operative motor deficits. This is due to the temporal summation of desynchronized D-waves at the segmental level. This desynchronization occurs in the conduction of the fast axons of the CST, which conduct D-waves at different speeds over the tumour site. In such cases, it is difficult to detect D-waves caudal to the lesion as D-waves might be of low amplitude or have a wider base, or even be completely absent [14].

## 3. Methodology for Intraoperative D-Wave Monitoring

Transcranial electrical stimulation (TES) is performed using constant current via corkscrew electrodes placed on the scalp according to the 10–20 International System [25]. These electrodes are positioned at C1/C2 or C3/C4. Cz − 1 cm/Cz + 6 cm can be also used. Of these stimulation montages, C1/C2 is preferred as it is associated with fewer stimulus artefacts and minimal patient movement.

Obtaining D-wave recordings requires a single pulse stimulation. The descriptions in the literature of the methods to elicit D-waves vary slightly. With constant current stimulators, stimulation usually consists of a single pulse with a pulse width of 0.5 ms [10,26]. Proposed recording parameters use a time base of 10–20 ms, from a single recording up to 5–20 averages, high-pass filters of 0.2–2 Hz (although some reports recommend up to 500 Hz), and low-pass filters of 1500–3000 Hz [10]. In our institution, we average four consecutive pulses of alternating polarity, with a frequency of stimulation of 0.5 Hz, a time base of 20 ms, a gain of 10 mV peak-to-peak, a high-pass filter of 100 Hz, and low-pass filter of 1000 Hz. We initially set the sensitivity to 100 µV per screen division, and we use an intensity slightly above the level needed to elicit muscle MEPs in all four limbs. However, we adjust these parameters whenever needed.

The single pulse stimulation technique for D-waves allows the continuous monitoring with real-time feedback to the surgical team without the inconvenience of patient movement. This is an advantage compared to muscle MEPs, which require a short train of (3 to 7) stimuli and thus may cause patient movement.

For recording, a special type of semi-rigid catheter electrode is used (Figure 1), which generally has 2 or 3 contacts separated by 18 mm each. The distance between contacts can vary slightly depending on the manufacturer. This catheter is positioned caudally in the subdural or epidural space [27,28]. Epidural positioning can be used in cases of swelling of the spinal cord, where there is fibrotic scar tissue from a previous surgery or when there is resistance to subdural placement. However, positioning in the subdural space reduces impedance and consequently results in a higher D-wave amplitude. Ideally, another electrode could be positioned cranially for a control in case of significant changes. It is recommended to secure the electrode to the adjacent tissue to prevent displacement during the procedure.

The D-wave has a triphasic morphology with a latency and amplitude that are dependent on the recording location (Figure 2C). At the cervical level, the peak latency is generally 3 to 6 ms, while at the thoracic level, it is 6 to 9 ms. The latency further depends on the presence or absence of conduction block secondary to the patient’s pathology, especially in large IMSCT. This can result in a decrease in the amplitude and an increase in the latency due to different conduction velocities of the CST fibres (desynchronization of the D-wave, as described above) [29].

Baseline recordings should be obtained before starting the resection of the lesion. To obtain a D-wave with maximal amplitude, stimulation intensity is increased stepwise until a further increase in stimulus intensity results in no further increase of amplitude (supramaximal threshold). As a rule of thumb, this is the intensity needed to obtain the MEPs of the lower limb muscles. Usually, a few averages are needed. By averaging 4–10 stimuli of alternating polarity, the signal-to-noise ratio can be improved. The relevant measurements of the D-wave are its presence and the peak-to-peak amplitude.

The monitoring of the D-wave begins immediately after exposure of the dura (epidural placement) or after opening of the dura and before myelotomy (subdural placement). The peak-to-peak amplitude is the primary parameter considered when evaluating significant changes in the D-wave value. A deterioration of 50% of the D-wave amplitude has been regarded as a warning criterion in IMSCT surgery and should not be exceeded [10,16,26]. The preservation of D-wave amplitude or a decrement of less than 50% is associated with a good long-term motor outcome. Conversely, a decrement of more than 50% in amplitude, or the total disappearance of the D-wave, is indicative of substantial injury to the CST and is therefore associated with permanent motor deficit [14,16,26,27,30]. However, in the case of intracranial air or caudal electrode displacement, a false positive alarm might be raised. It is in these cases that the positioning of a proximal catheter becomes beneficial [31].

These warning criteria have been intensively investigated in the context of IMSCT surgery. They can also be applied in selected cases of intradural extramedullary tumours and surgery for syringomyelia. However, their applicability in scoliosis correction and other orthopaedic procedures has been questioned [28]. This might be because, during correction, spinal cord rotation within the spinal canal occurs, affecting the relative position of the epidural recording catheter to the CST. This can lead to overestimation of the amplitude decrement of the D-wave, leading to a false positive result and a premature cessation of the surgery [31].

## 4. D-Waves in Paediatric Spinal Cord Surgery

The principles of IOM are the same in children as in adults, but some special considerations apply to paediatric patients.

At the time of birth, synapses between the CST and the lower motor neuron already exist, likely initiated in the last trimester of gestation. The myelination of the CST begins at birth and continues until approximately 24 months of age or even longer [32]. In this context, it has been demonstrated that central motor conduction time is affected by age. It is longer in newborns than in adults, reaching adult values in the upper limbs at 2 to 4 years and in the lower limbs at 6 years [33].

Compared to adults, children under 10 years have higher MEP threshold values. This has been investigated using transcranial magnetic stimulation. It is also known that the stimulation threshold for TES has a negative correlation with age [34]. When performing TES in the paediatric population, the clinician should consider the anatomical disposition of the motor cortex when placing the stimulation electrodes. Rivet et al. [35] demonstrated that the distance of the primary motor area moves by 1.5 mm every year in the anteroposterior direction from 2 months to approximately 8 years. They suggest a more anterior placement of electrodes in the head, modifying the classical 10–20 International System [25] from C3/C4 to FC3/FC4 or C1/C2 + 1 cm. In patients younger than 18 months and those with an open fontanelle, the placement of stimulation electrodes should be adapted accordingly.

The D-wave might be difficult to record in children younger than 36 months, in whom myelination is still incomplete [31,35]. This, combined with the presence of structural alterations such as an IMSCT, makes it even more difficult to record reliable D-waves. Szelényi et al. studied 19 children under 3 years with IMSCT. The authors obtained D-wave recordings in seven cases, concluding that CST maturation is sufficient for the epidural recording of a D-wave from the thoracic spinal cord. They could not record a D-wave in children aged under 21 months in their series [36]. However, another case report presented D-wave recordings made in a 10-month-old child [37].

Despite these challenges, D-wave monitoring has been demonstrated to be valuable in paediatric patients with IMSCT. A clear example is a recent study by Antkowiak et al. [38], which included 23 children undergoing IMSCT surgery with IOM. The D-wave was measurable in 60.9% of cases and exhibited a specificity of 92.3% and a sensitivity of 100% for predicting postoperative motor deficits. The authors reported that IOM did not limit the extent of tumour resection, with similar GTR rates achieved in patients with and without IOM alerts. Those findings are further supported by a larger series of 100 adult patients with intramedullary ependymomas reported by Skrap et al. In their study, GTR was achieved in 89% of ependymoma surgeries, and long-term functional independence was maintained in 82% of patients. D-waves adequately predicting long-term functional motor outcomes were successfully monitored in 67% of cases [17].

Recently, we reported on a cohort of five paediatric patients in which, besides D-wave monitoring, we introduced a specific mapping technique to localize the CST [39]. Guiding the surgical resection of IMSCT by mapping techniques might be a useful additional neurophysiological tool; however, this is beyond the scope of the current article.

## 5. Future Perspectives on CST Monitoring: The Anti D-Wave

Since 1986, many authors have applied electrical stimulation of the spinal cord to record signals on the scalp; these are called spinal cord evoked potentials (ScEPs) [40,41]. Two distinct categories of evoked potentials have been identified: an early potential and a late potential. Interestingly, most researchers have focused their investigations on the potential with the longer latency, corresponding to the evoked electrical activity originating from the dorsal columns. However, an intriguing hypothesis was that the early component might be linked to the CST [42].

In 2016, Costa et al. embarked on a quest to explore the early potential (called the anti D-wave), evoked by epidural stimulation of the spinal cord in both neurologically compromised and uncompromised patient populations [43]. This study had several objectives: to delineate the origin and spatial distribution of the signal, analyse its response to varying stimulation parameters, document the recorded metrics, and establish correlations with D-wave behaviours. They performed electrical epidural stimulation via D-wave catheter electrodes both cranially and caudally to the surgical site. The stimulation comprised a single stimulus of 300 μs, delivered at a rate of 3 Hz, with an intensity typically ranging from 10 to 30 mA. For recording purposes, corkscrew electrodes were positioned on the scalp along the midline, over Fz, Cz′ and Pz referenced to the electrode Fpz from the 10–20 International System [25]. The time base was 50–100 ms. To capture the maximal cortical response, they averaged 30 to 100 recordings.

The result was an early potential identified as the “anti D-wave”, presumed to be non-synaptic in nature, as evidenced by its resistance to stimulation at very high rates. This wave, characterized by a triphasic pattern, exhibits a marginally longer latency compared to the D-wave within the same patient (0.8 ± 0.48 ms). This discrepancy is likely attributable to a deeper activation of the CST when the D-wave is elicited via TES from the scalp. Furthermore, the observed latency shift between the D-wave and the anti D-wave aligns with the hypothesis suggesting that the anti D-wave is generated by the CST. The features of the anti D-wave exhibit a close correlation with those of the D-wave, particularly concerning its distribution, response to filtering, stimulus rate, and absence in paraplegic patients. However, further studies are necessary to evaluate the usefulness of the anti D-wave as a monitoring method.

## 6. Illustrative Case of Feasibility of D-Wave and Anti D-Wave Recordings

A 41-year-old woman presented with gait ataxia and severe pain in her lower limbs, predominantly affecting the right lower limb. She was diagnosed with an intraspinal extradural lesion at the Th1–Th2 level (Figure 2A), and surgery was planned. Intraoperatively, before starting tumour resection, a D-wave catheter electrode of two contacts separated by 18 mm each other was placed distally into the subdural space. Initial D-wave recordings were obtained by stimulating C3/C4, resulting in a triphasic potential with a peak latency of 5.3 ms and a peak-to-peak amplitude of 24.6 uV (Figure 2C).

Following this, bipolar stimulation was performed via the two-contact catheter electrode with a current intensity of 20 mA, a frequency of 1.7 Hz, a pulse width of 0.5 ms and a cathodal polarity (the cathode was located proximal and the anode, distal). Recording channels, as seen in Figure 2D, included sagittal Cz′/Fz, interhemispheric C3′/C4′ and C3/C4 montages (the same as used to elicit the D-wave). We averaged thirty recordings. There was minor movement of the surgical field during the stimulation, which did not interfere with the surgical manoeuvre. Since the anti D-wave has a slightly longer latency than the D-wave (with a peak latency of 5.3 ms in the case presented), responses recorded with a peak latency around 6 ms were identified as the anti D-wave. Additionally, a wave with longer latency and higher amplitude, likely corresponding to the SSEP elicited by the direct stimulation of the dorsal columns, was observed.

Costa et al. 2016 [43] report that stimulation of the spinal cord at the cervical or cervico-thoracic level results in negative waveforms of the anti D-wave and the somatosensory evoked potential. This is likely due to the stimulation of the corticospinal tract of the upper limbs. Our recordings are consistent with these findings. In the case described, stimulation was performed at the upper thoracic level with an intensity of 20 mA. This likely caused the current to spread to the corticospinal tract fibres of the upper limbs, resulting in the observed negative polarities. Further, mild differences in latencies of the anti D-wave in the different recording montages were not surprising. These findings are similar to clinical observations for somatosensory evoked potential recordings.

Postoperative magnetic resonance imaging (Figure 2B) demonstrated the complete resection of the intraspinal component of a histopathologically confirmed chordoma.

## 7. Conclusions

D-wave monitoring is a commonly used technique in IOM, which may reliably guide the resection of IMSCT and predict long-term motor outcomes. The potential role of anti D-waves needs to be investigated in future studies.

## Figures and Tables

**Figure 1 medicina-60-01242-f001:**
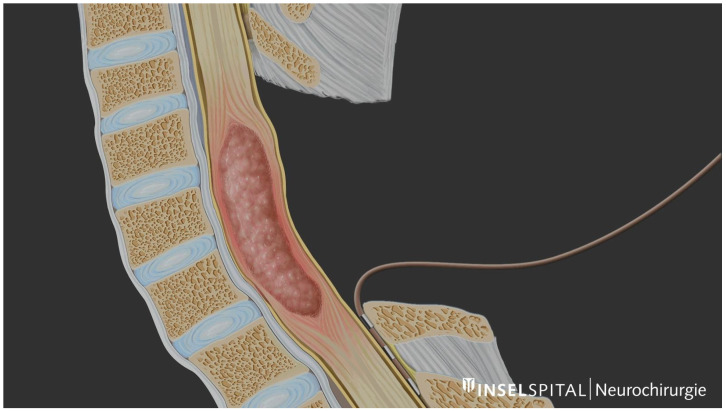
Schematic illustration of distal D-wave catheter placement below the level of the tumour. © Inselspital, Bern University Hospital, Dept. of Neurosurgery.

**Figure 2 medicina-60-01242-f002:**
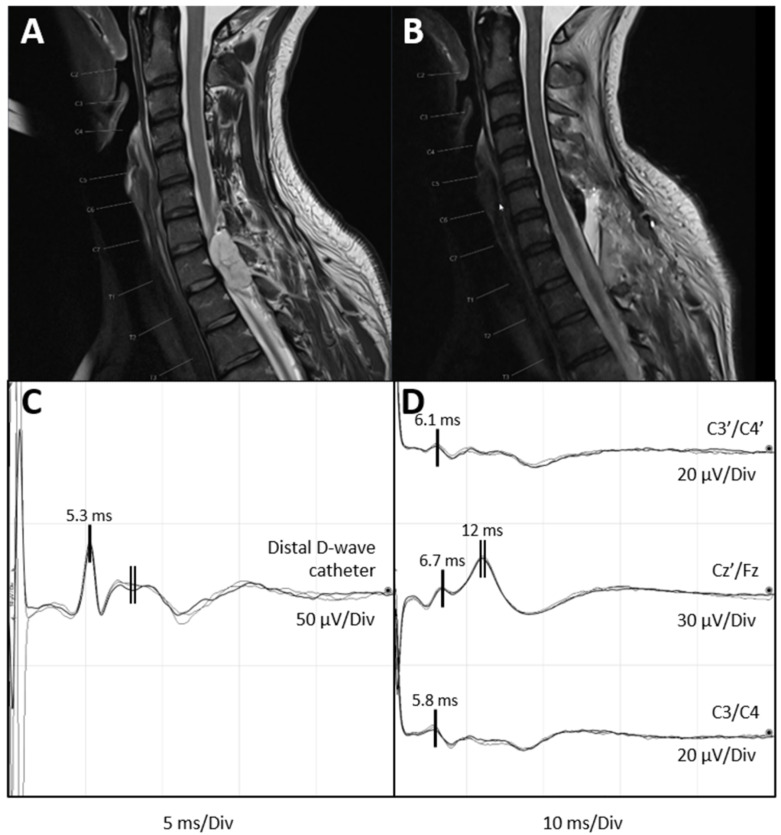
Preoperative and postoperative MRI scans and intraoperative neurophysiological monitoring recordings. (**A**) Preoperative MRI (T2-weighted image, sagittal plane) showing the intradural extramedullary chordoma at the level of Th1–Th2. (**B**) Postoperative MRI confirming tumour resection. (**C**) Intraoperative D-wave recording distal to the tumour site at the level of Th2–Th5 during tumour removal. The D-wave is characterized by a short-latency triphasic wave (single vertical line). Peak latency: 5.3 ms. Amplitude: 24.7 μV. Note the later artefact of lower amplitude and longer duration caused by paraspinal muscles (double vertical line). It should not be mistaken as a D-wave. (**D**) Anti D-wave recording illustrating a short latency potential with a peak latency ranging from 5.8 ms to 6.7 ms, depending on the recording channel (single vertical lines), followed by a higher amplitude and longer duration potential with a peak latency of 12 ms observed in the channel Cz′/Fz (double vertical line). This later potential corresponds to the somatosensory evoked response (right vertical line). Recordings were obtained at C3′/C4′, Cz′/Fz, and C3/C4 of the 10–20 International System. In C3/C4 (corresponding to the montage over motor cortex), and the anti D-wave (representing the antidromic activity of the CST) reaches its hsighest amplitude (1.8 μV). The highest amplitude of the somatosensory response is recorded in the sagittal montage Cz′/Fz (midline, frontoparietal). Sensitivity is tailored independently in different channels for visualization purposes. ***CST:*** corticospinal tract. ***MRI:*** magnetic resonance imaging.

## Data Availability

The data that support the findings of this study are available from the corresponding author upon reasonable request.

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
