# Peer review of "A Concise Guide to D-Wave Monitoring during Intramedullary Spinal Cord Tumour Surgery"

_medicina, 2024, doi:10.3390/medicina60081242_

Round 1

Reviewer 1 Report

Comments and Suggestions for Authors

The authors present a work that reasonably summarizes the use of D-wave monitoring in intramedullary surgery. They briefly review the subject's current state (most important references) and include some practical recommendations that may be useful to readers. The article is written clearly and well-structured.

Below are some commentaries.

Major

1.- Figure 2, panel D: The authors marked as anti-D wave a negative peak in the three derivations used for recording. However, in the series published by Costa et al. in 2016 (cited by the authors), the anti-D-wave exhibited a positive polarity when recorded over the scalp using anterior montages (Fz–Fpz and  Cz′–Fpz), and only the signal had a negative peak (specular image) when recorded at Pz–Cz′. Costal et al. didn't explore interhemispheric derivations for the recording. Still, they stimulated the spinal cord with the same catheter at the thoracic level and recorded an opposite polarity in the midline frontoparietal channel (Cz'/Fz or Cz'/Fpz) over the scalp. How do the authors explain that?

Minor

1.- In the methodology for intraoperative D-wave monitoring, in the second paragraph, when they explain the parameters used in their institution, what does "a gain of 10mV peak-to-peak" mean? Also, consider rephrasing the sentence "We start with a sensitivity of 100". 

2. In the third paragraph of the methodology for intraoperative D-wave monitoring, the authors say: " compared to muscle MEPs, which require pulse train stimulation." Do they refer to the multi-pulses/short-train technique? "Pulse train stimulation" seems to be vague. 

3.- In the illustrative case: Does bilateral C3´/C4´ refer to interhemisferic? Interhemisferic or coronal may be more appropiate.

4. In the illustrative case, the authors say they used cathodal polarity for the stimulation, but the specific electrodes used are missing. The authors indicate a stimulus frequency of 1.7 Hz. It can be deduced that they were using simple repetitive pulses for stimulation. However, how many traces were averaged in the acquisition?

5.- Did direct spinal cord stimulation at 20 mA generate movement in the patient or the surgical field? If so, could the movement in the surgical field be a limitation to using the anti-D-wave as a continuous monitoring technique? 

6.-The figure 2 shows the postoperative MRI; however, the final D-wave and anti-D wave traces are not included. What were the D and anti-D waves like at the end of the tumor resection? Were there any intraoperative events?

7.-In Figure 2, panel D: The left vertical line indicates the peak latency of the anti-D wave, according to the authors. However, looking at the traces in attention, you can see the maximum peak does not have the same latency in the three derivations: it coincides with the line marked in the first recording channel (C3'/C4'), it is slightly delayed in the second channel (Cz'/Fz) and is slightly advanced in the last one. How do the authors explain this fact? If different recording derivations cause slight latency variations, a different way of marking the potential in the figure should be considered. Instead of a line, individual markers on each path might be more appropriate and clear for the readers.

Author Response

Major

1.- Figure 2, panel D: The authors marked as anti-D wave a negative peak in the three derivations used for recording. However, in the series published by Costa et al. in 2016 (cited by the authors), the anti-D-wave exhibited a positive polarity when recorded over the scalp using anterior montages (Fz–Fpz and  Cz′–Fpz), and only the signal had a negative peak (specular image) when recorded at Pz–Cz′. Costal et al. didn't explore interhemispheric derivations for the recording. Still, they stimulated the spinal cord with the same catheter at the thoracic level and recorded an opposite polarity in the midline frontoparietal channel (Cz'/Fz or Cz'/Fpz) over the scalp. How do the authors explain that?

Response: In the referred paper by Costa et al. 2016, section 3.3, the authors report that stimulation of the spinal cord at the cervical or cervico-thoracic level results in negative waveforms of the anti D-wave and the somatosensory evoked potential. This is likely due to the stimulation of the corticospinal tract of the upper limbs. Our recordings are consistent with these findings. In the case described in our paper, stimulation was performed at the upper thoracic level with an intensity of 20 mA. This likely caused the current to spread to the corticospinal tract fibres of the upper limbs, resulting in the observed negative polarities.

Minor

1.- In the methodology for intraoperative D-wave monitoring, in the second paragraph, when they explain the parameters used in their institution, what does "a gain of 10mV peak-to-peak" mean? Also, consider rephrasing the sentence "We start with a sensitivity of 100".

Response: The gain is the degree to which amplification increases signal power. It represents how much an amplifier increases the strength of the input signal. A gain of 10 mV peak-to-peak means that the amplifier increases the input signal so that the difference between the highest and lowest points of the output signal is 10 milivolts [1].

We rephrased the suggested sentence (lines 139 and 140).

  1. In the third paragraph of the methodology for intraoperative D-wave monitoring, the authors say: " compared to muscle MEPs, which require pulse train stimulation." Do they refer to the multi-pulses/short-train technique? "Pulse train stimulation" seems to be vague. 

Response: We rephrased the term according to the advice of the reviewer (lines 144 and 145).

3.- In the illustrative case: Does bilateral C3´/C4´ refer to interhemisferic? Interhemisferic or coronal may be more appropiate.

Response: We appreciate the commentary of the reviewer and agree that probably “bilateral” is not an appropriate term. We rephrased it (line 295).

  1. In the illustrative case, the authors say they used cathodal polarity for the stimulation, but the specific electrodes used are missing. The authors indicate a stimulus frequency of 1.7 Hz. It can be deduced that they were using simple repetitive pulses for stimulation. However, how many traces were averaged in the acquisition?

Response: We specified the number of contacts of the catheter used, the type of stimulation and the amount of averages (lines 288, 289, 292, 294, and 296).

5.- Did direct spinal cord stimulation at 20 mA generate movement in the patient or the surgical field? If so, could the movement in the surgical field be a limitation to using the anti-D-wave as a continuous monitoring technique? 

Response: Stimulation did cause minor movement, which was not interfering with the surgical manoeuvres. This is now clarified in the text (lines 297 and 298).

6.-The figure 2 shows the postoperative MRI; however, the final D-wave and anti-D wave traces are not included. What were the D and anti-D waves like at the end of the tumor resection? Were there any intraoperative events?

Response: There were no intraoperative events during the surgery. Our aim was to present D-wave and anti D-wave recordings to provide readers with examples of the expected waveforms.

Therefore, we did not include pre- and post-operative D-wave recordings, as they were identical.

Regarding the anti D-wave, its clinical use is still not validated. Thus, we performed the stimulation for  a "proof of concept/feasibility approach" at our centre. The recordings shown were taken during one surgical time point but not for continuous monitoring.

7.-In Figure 2, panel D: The left vertical line indicates the peak latency of the anti-D wave, according to the authors. However, looking at the traces in attention, you can see the maximum peak does not have the same latency in the three derivations: it coincides with the line marked in the first recording channel (C3'/C4'), it is slightly delayed in the second channel (Cz'/Fz) and is slightly advanced in the last one. How do the authors explain this fact? If different recording derivations cause slight latency variations, a different way of marking the potential in the figure should be considered. Instead of a line, individual markers on each path might be more appropriate and clear for the readers.

Response: We are not surprised of mild differences in latencies of the anti D-wave in the different recording montages. Similar to clinical observations for somatosensory evoked potential recordings, we have already found these slight latency variations in the anti D-wave.

Nevertheless, we agree with the suggestion of the reviewer to specify the latency of the anti D-wave in the different recording channels. It was modified in the figure. Further we added an explanation to the legend of figure 2 (lines 171 to 177).

Reviewer 2 Report

Comments and Suggestions for Authors

This is a narrative report along with a short case report. 

Line 13: not exclusively for intramedullary spinal cord tumours, but also for extrameducllary tumors, cord detethering etc

The authors need to define what they mean by muscle MEP. They also need to provide a brief description of what MEP and what SSEP is.

Page 2 line 64: what do they mean by threshold activation? why is the brain mentioned here? They need to explain things for readers who do not have the background knowledge

Line 88-90: what is the diffenrence between post—op motor function and long term outcome? aren’t these two closely related? if not, explain

An explanation of the exact pathway of the D-waves is needed. 

line 122: what is the denominator “div”?

Line 150: chordomas are epidural tumors. If that specific tumor had eroded the dura, then it should be described as that.

Author Response

Line 13: not exclusively for intramedullary spinal cord tumours, but also for extrameducllary tumors, cord detethering etc

Response: Classical D-wave monitoring is validated for intramedullary spinal cord tumours. As explained in the last paragraph of our manuscript (section 3, “Methodology for intraoperative D-wave monitoring”), D-wave monitoring can also be applied in selected cases of intradural extramedullary tumours and surgery for syringomyelia (added in the text, lines 13 and 14). However, its applicability in scoliosis correction and other orthopaedic procedures has been questioned.  During correction, spinal cord rotation within the spinal canal occurs, affecting the relative position of the epidural recording catheter to the CST. This can lead to overestimation of the amplitude decrement of the D-wave, leading to a false positive result and a premature cessation of the surgery.

The authors need to define what they mean by muscle MEP. They also need to provide a brief description of what MEP and what SSEP is.

Response: According to the reviewer’s suggestion we added those aspects to the first paragraph of the introduction (lines 25-27 and 35-38).

Page 2 line 64: what do they mean by threshold activation? why is the brain mentioned here? They need to explain things for readers who do not have the background knowledge

Response: The paragraph was modified to gain clarity (lines 72 to 77).

Line 88-90: what is the diffenrence between post—op motor function and long term outcome? aren’t these two closely related? if not, explain

Response:  To assess motor outcome of the patients after surgery two time points might be important in the clinical and especially oncological context. First, motor outcome directly after surgery. This is mostly reflected by preserved muscle MEPs. Second, the potential to recover, which means that the patient wakes up with a motor deficit directly after surgery but recovers to preoperative motor function over time (transient deficits). This might be acceptable in a context of resection of an IMSCT as the patient might undergo rehabilitation. The potential to recover is reflected by a preserved D-wave (even in case muscle MEPs are lost) [1-4] .

An explanation of the exact pathway of the D-waves is needed. 

Response: It is an asynaptic potential, elicited by electrical stimulation of the subcortical white-matter of the brain, reflecting the first motor neuron, and resulting from direct activation of fast-conducting, thickly myelinated corticospinal tract fibres. We had explained this aspect in section 2 (Neurophysiological background), paragraphs 1 and 2 (lines 70 to 87).

line 122: what is the denominator “div”?

Response: It means “Division”. In the context of this paragraph, it means that the distance between two horizontal lines in the screen (corresponding to one division) displays an amplitude of the response equivalent to 100 µV. We rephrased the paragraph accordingly (line 139).

Line 150: Chordomas are epidural tumors. If that specific tumor had eroded the dura, then it should be described as that.

There was no infiltration of the dura, the case presented was used as a feasibility case to obtain anti- D-wave recordings.
